# Feeding Dairy Ewes with Fresh or Dehydrated Sulla (*Sulla coronarium* L.) Forage. 1. Effects on Feed Utilization, Milk Production, and Oxidative Status

**DOI:** 10.3390/ani12182317

**Published:** 2022-09-06

**Authors:** Riccardo Gannuscio, Marialetizia Ponte, Antonino Di Grigoli, Giuseppe Maniaci, Adriana Di Trana, Monica Bacchi, Marco Alabiso, Adriana Bonanno, Massimo Todaro

**Affiliations:** 1Dipartimento Scienze Agrarie, Alimentari e Forestali (SAAF), Università Degli Studi di Palermo (UNIPA), 90128 Palermo, Italy; 2Scuola di Scienze Agrarie, Forestali, Alimentari ed Ambientali (SAFE), Università Degli Studi Della Basilicata, 85100 Potenza, Italy; 3Dipartimento Agraria, Università Mediterranea di Reggio Calabria (UNIRC), 89122 Reggio Calabria, Italy

**Keywords:** dehydrated sulla forage, condensed tannins, dairy ewes, feed intake, digestibility, plasma oxidative balance

## Abstract

**Simple Summary:**

Sulla forage, containing condensed tannins, is renowned in the Mediterranean area for its excellent chemical composition and agronomic traits, the suitability for grazing or hay production, and favorable impact on ruminants’ productivity. Condensed tannins of sulla forage exert various positive effects, because they contribute to reducing methanogenesis in the rumen, protect dietary proteins from rumen degradation, limit the ruminal biohydrogenation of polyunsaturated fatty acids, and ensure animal health by anthelmintic action. Moreover, the antioxidant activity of condensed tannins may contribute to improving the antioxidant status of animals, as well as the oxidative stability and antioxidant properties of animal products, with important health benefits for human consumers. The conservation methods of sulla forage, such as dehydration and the hay-making process, can influence the content and bioavailability of condensed tannins. In this research, the performance of dairy ewes fed with pelleted dehydrated sulla forage, in terms of feed intake, digestibility, milk production, and oxidative status, were compared with those of ewes fed diets based on sulla hay and fresh sulla forage. The results confirmed the positive effects of sulla forage, also when fresh sulla forage is replaced with dehydrated forage. On this basis, the storage and use of sulla dehydrated pellets represent a valid opportunity to exploit the potential of sulla forage in periods of low availability of grazing sources.

**Abstract:**

Feeding dairy ewes with fresh sulla forage (FSF), a legume species containing condensed tannins (CT), has been shown to increase feed intake, milk yield, and casein and enhances the oxidative status of animals. Dehydration of FSF could be an alternative to hay-making to preserve the nutritional properties. This research aimed to compare the responses of dairy ewes fed with diets based on sulla hay (SH), pelleted dehydrated sulla forage (DSF), or FSF in terms of efficiency of feed utilization, milk production, and the balance between oxidant (reactive oxygen metabolites (ROMs)) and antioxidant (biological antioxidant potential (BAP)) substances at the plasma level. Ten first-lambing (FL) and ten third-lambing (TL) ewes of the Valle del Belice breed at 60 days in milk were allocated into five homogeneous groups fed with different diets in a partial 5 × 2 Latin square design with two phases. The diets differed for the forage basis: SHL = SH ad libitum; DSF2 = 2 kg/day DSF per head plus SH ad libitum; FSF2 = 2 kg/day FSF per head plus SH ad libitum; FSF4 = 4 kg/day FSF per head plus SH ad libitum; FSFL = FSF ad libitum. A commercial concentrate was provided to FL (0.8 kg/day per head) and TL (1.2 kg/day per head) ewes. Dehydration induced slight variations in the content of protein and fiber, showed no loss of vitamin E and polyunsaturated fatty acids, and decreased the level of CT and polyphenols. The DSF2 diet resulted in a higher intake of dry matter, protein, and vitamin E compared to the other diets, whereas, compared to the FSFL diet, its intake was analogous for net energy and was lower in CT and polyphenols. The DSF2 diet was comparable to FSF4 and FSFL diets for milk yield, and to all diets for casein content and the clotting ability of milk. Ewes fed a DSF2 diet exhibited lowest values of ROMs and oxidative stress index (OSI = ROMs/BAP), indicating a better oxidative status, presumably due to the antioxidant protection exerted by the higher vitamin E intake and CT metabolites. These results confirmed the positive effects of FSF on milk production, especially due to CT intake, in improving the efficiency of dietary protein utilization, and showed how favorable effects also occur when FSF is replaced by DSF instead of SH.

## 1. Introduction

Sulla (*Sulla coronarium* L.) is a biennial forage legume species, typical of cereal-based crop rotation in Mediterranean areas where it is recognized for its excellent agronomic traits, suitability for grazing, and silage or hay production [1], and the favorable impact of both fresh and preserved forage on ruminants’ productivity, particularly related to its composition. Indeed, sulla forage is characterized by high protein, low fiber, and a relatively high ratio of non-structural to structural carbohydrates that improves its palatability, animal intake, and digestibility [2,3,4]. Sulla also shows an interesting content of total phenolic compounds, consisting mainly of condensed tannins (CT), known also as proanthocyanidins [5,6,7].

Condensed tannins are secondary plant metabolites acting as natural agents of several biological activities in the rumen [8]. Indeed, CT contribute to reducing methanogenesis, mitigating methane emissions [6,9] and, due to their ability to bind with proteins, protect dietary proteins from rumen degradation, thus enhancing the efficiency of protein utilization in ruminants [6,10,11] and reducing nitrogen excretion into the environment [3]. Moreover, limiting the ruminal biohydrogenation of polyunsaturated fatty acids (PUFA), CT also improve the fatty acid (FA) profile of animal products by increasing the concentration of PUFA beneficial for human health [12]. In addition, CT ensure animal welfare and health because they exert an anthelmintic effect that is able to control gastrointestinal nematode parasites [13]. They also possess antioxidant activity that improves the antioxidant status of animals, as well as the oxidative stability and antioxidant properties of products [8,14,15], which also provides important health benefits for human consumers.

In Mediterranean environments, sulla forage showed CT levels ranging from 8 to 50 g/kg of the whole plant dry matter (DM) [16] that correspond to moderate contents (<60 g/kg DM) [17]. At these moderate concentrations, CT do not provoke any detrimental effect on animal feed intake, digestibility, or productivity. They are also able to bind with proteins and reduce the rumen proteolysis, thus enhancing the proteins bypassing the rumen to be digested and absorbed in the intestinal tract, improving the dietary protein utilization [6,11,18]. Indeed, CT of sulla forage contributed to improving the performance of goats [19,20] and dairy ewes [3,4] in terms of efficiency of feed utilization, milk yield, and milk casein level. The CT contained in sulla forage were also able to inhibit the complete biohydrogenation of dietary PUFA in the rumen, thus increasing the level of health PUFA in dairy products from small ruminants [4,20,21]. In addition, the CT ingested by goats fed fresh sulla forage improved the plasma oxidative status of animals and increased the polyphenol content and the total antioxidant capacity of milk [14], suggesting a certain systemic bioavailability and, in accordance with Gladine et al. [22], a possible transfer of dietary CT to plasma and milk. Indeed, despite some conflicting results, it is widely accepted that, depending on the complexity of their polymeric structure and the rumen microbial degradation, CT escaping from rumen are metabolized along the intestine, and their metabolites of low molecular weight are absorbed into circulation to be deposited into various tissues, where they can exert a direct antioxidant action [8,23]. However, both dehydration and the hay-making process can influence the content, the structure, and the bioavailability of CT in sulla forage; indeed, heat treatments seem to reduce the polyphenolic components of vegetables but improve their bioaccessibility [24,25].

On this basis, the advantageous effects of feeding fresh sulla forage observed on ruminants’ performance and the technological, nutritional, and health properties of their dairy products are strictly linked to the contents of nutrients, polyphenols, and CT in this forage species. Consequently, to exploit sulla forage and its properties in periods of low availability, it is important to implement techniques and procedures to stabilize the significant nutrients and bioactive molecules in the conserved forage. In this regard, a valid solution to preserve the properties of sulla forage can be represented by dehydration, followed by pelleting, as an alternative to the hay-making process, which causes excessive losses of biomass and its components.

Thus, the aim of this research was to compare the performance and dairy products of ewes fed with pelleted dehydrated sulla forage in their diet with those of ewes fed with diets based on sulla hay and fresh sulla forage at different levels. The results of this study were divided into two parts and are presented in two different papers: this first part reports the responses of dairy ewes in terms of feed intake, digestibility, milk production, and oxidative status, evaluated in the function of the diets, whereas the effects on technological, nutritional, and health properties of sheep cheeses, including FA profile, are reported in the second part and are the subject of another paper [26].

## 2. Materials and Methods

### 2.1. Animals, Experimental Design, and Feeding Treatments

The experiment was carried out at a commercial farm located in the territory of Santa Margherita di Belìce, in the province of Agrigento (Sicily, Italy), during a six-week period from April to May. A total of 20 ewes of the Valle del Belìce breed were selected from a farm batch of ewes at 60 days in milk, and randomly divided into five experimental groups that were homogeneous for parity (10 first- and 10 third-lambing ewes), live weight (48.86 ± 5.85 kg), and milk yield (2101 ± 407 g/day). A few days after selection and for the entire experimental period, the ewes were housed in pairs in 10 wide, straw-bedded pens placed indoors and equipped with a feeder and water trough; thus, each group was composed of two pairs, one of first-lambing (FL) and the other one of third-lambing (TL) ewes. Firstly, the groups were submitted to a 10-day period of adaptation to the new housing condition, during which the ewes were fed a preliminary diet based on sulla hay ad libitum and a commercial concentrate feed (CCF), supplied at the level of 800 g/day for FL or 1200 g/day for TL, according to their daily requirements [27]. Then, each group was fed one of five experimental diets, according to a partial 5 × 2 Latin square design with two phases, each composed of 10 days for adaptation to diets and 5 days for measuring and sampling. The diets were different for the type (hay, dehydrated, fresh) and level of sulla forage: SHL = sulla hay (SH) ad libitum; DSF2 = 2 kg/day dehydrated sulla forage (DSF) per head plus SH ad libitum; FSF2 = 2 kg/day fresh sulla forage (FSF) per head plus SH ad libitum; FSF4 = 4 kg/day FSF per head plus SH ad libitum; FSFL = FSF ad libitum. The diets were integrated with the same CCF used in the adaptation period, supplied at levels equal to 800 g/day per head for PR and 1200 g/day per head for PL, partitioned in two daily meals, which were always entirely consumed by the ewes. The FSF was mowed daily in the morning from a close sward and cut roughly. To produce the DSF, sulla forage was harvested at the phenological stage of full flowering, cut at 3–4 cm, dehydrated in a small pilot-plant working with diesel fuel, under 50 °C hot air for 18–24 h until reaching approximately 15% humidity, and then pelleted. The daily rations of FSF and DSF were divided into two meals supplied after morning (7:00) and afternoon milking (15:00); the SH and FSF offered ad libitum were supplied at the same times of day, in amounts to maintain enough refusals.

The experiment protocol had the approval of the Animal Welfare Body of the University of Palermo (2021-UNPA-CLE-0059470) who ruled as not applicable the requirements established by the National Legislative Decree n. 26/2014, implementing the Directive 2010/63/EU.

### 2.2. Measurements and Sampling

The live weight of ewes was recorded at the start and end of each experimental phase. Other measurements and sampling were performed during the final 5 days of each phase. The offered SH, DSF, FSF, and CCF, as well as the corresponding refused amounts of each pen, were weighed daily; the offered resources and the refused SH and FSF were sampled three times to calculate the amount and composition of dietary intake. Both offered and refused FSF samples were freeze-dried before analysis. Individual milk yield was recorded daily at morning and afternoon milking and sampled twice. To assess the oxidative status of ewes, individual blood was sampled from all ewes at the end of each experimental phase, collecting a total of 40 samples; blood samples were taken in the morning from each fasted ewe by jugular venipuncture, using two vacutainer tubes, one containing lithium heparin to obtain plasma, and the other without anticoagulant for serum. Samples for plasma were immediately placed on ice and then centrifuged at 3000 rpm for 10 min at 4 °C, whereas samples for serum were maintained at room temperature until clotting, and then centrifuged at 3000 rpm for 7 min; both plasma and serum samples were stored at −80 °C until analysis. At the end of each experimental phase, individual feces samples were collected from ewes to estimate dietary digestibility; fecal grabs were taken in the morning, directly from the rectum, then freeze-dried for the successive analyses.

### 2.3. Analyses

Samples of offered and refused forages and those of CCF were analyzed according to AOAC [28] procedures to determine dry matter (DM) (method 934.01), crude protein (CP) (method 2001.11), ether extract (method 920.39), and ash (method 942.05). The fibrous fractions, as neutral detergent fiber using heat-stable amylase and exclusive of residual ash (aNDFom) (method 2002.04), ash free acid detergent fiber (ADFom) (method 973.18), and acid detergent lignin (ADL) (method 973.18), were determined in accordance with AOAC [28] and Van Soest et al. [29]. Non-structural carbohydrates (NSC, %) content was calculated as (100 − (CP% + EE% + ash% + aNDFom%)). The net energy for lactation (NEL, kcal/kg DM) of forages and CCF was estimated according to INRA [27].

The vitamin E (α-tocopherol) content in feed samples was determined by reversed phase HPLC methods, as reported by Panfili et al. [30] and Manzi et al. [31]. Briefly, aliquots (0.5 g) of feed were digested with 2 mL of KOH (60% aqueous solution, w/v), 2 mL of 95% ethanol, 1 mL of NaCl (1% aqueous solution, w/v), and 5 mL of an ethanolic solution of pyrogallol (6%, w/v) acting as an antioxidant. After digestion in a water bath at 70 °C, the suspension was cooled for 30 min, supplemented with 5 mL of NaCl solution (1%, w/v) to prevent emulsification, and then extracted with 10 mL of n-hexane/ethyl acetate (9:1, *v*/*v*). The lower aqueous layer was extracted three more times, with 5 mL of n-hexane/ethyl acetate (9:1, *v*/*v*) each time. The pooled organic layers were evaporated with a rotary evaporator at 30 °C, and the dry sample was dissolved in 3 mL of methanol for HPLC. A sample volume of 20 µL was injected using HPLC equipment, previously filtered using a 0.20 µm PTFE filter. All determinations were carried out in duplicate.

Extracts of the feeding resources were prepared to measure in duplicate their amounts of condensed tannins (CT), total polyphenols, and trolox equivalent antioxidant capacity (TEAC). Briefly, powdered, freeze-dried feed samples (0.75 g) were mixed with 25 mL acetone/water (70:30, *v*/*v*), sonicated in an ultrasonic water bath (LBS1 Sonicator; Falc Instruments, Treviglio, Italy) for 30 min at 30 °C, and then centrifuged at 6000 rpm at 4 °C for 15 min; the supernatants were filtered through Whatman No. 541 filter paper and kept at −18 °C until further analysis.

Extracted samples were analyzed for CT, expressed as delphinidin equivalent (g DE/kg DM) [32], using the butanol-HCl assay [33], total polyphenols, expressed as gallic acid equivalent (g GAE/kg DM), by the Folin-Ciocalteau colorimetric method [34], and TEAC (mmol trolox/kg DM), according to Re et al. [35], as described by Todaro et al. [36]; the absorbance of samples was read at 550, 725, and 734 nm for TC, polyphenols, and TEAC, respectively, using a HUCH DR3900 spectrophotometer (Hach, Loveland, CO, USA).

Freeze-dried feces samples were analyzed for determination of DM, CP, aNDFom, ADL, acid insoluble ash (AIA), ENL, CT, and polyphenols, as described for feeds, to estimate the digestibility using the AIA content in the feces as an internal marker, according to Sunvold and Cochran [37].

Daily milk samples collected from each ewe were analyzed for lactose, fat, protein, casein, and urea by infrared method (Combi-foss 6000, Foss Electric, Hillerød, Denmark). Individual milk samples were also evaluated for clotting ability using a Formagraph instrument (Foss Electric); coagulation time (r, min), curd-firming time (k_20_, min), curd firmness (a_30_, mm), and curd firmness after twice the clotting time (a_2r_, mm) were measured in 10 mL of milk at 35 °C with 0.2 mL diluted solution (1.6:100) of rennet (1:15,000; Chr. Hansen, Parma, Italy).

Plasma oxidants and antioxidant capacity were measured, respectively, as reactive oxygen metabolite-derived compounds (d-ROMs, expressed as Unit Carr), especially those consisted of hydroperoxides generated by the oxidation of biomolecules, and biological antioxidant potential (BAP, mmol/l reduced iron equivalent), measuring the plasma capacity to reduce iron from ferric (Fe^3+^) to ferrous (Fe^2+^) form, using two different kits from Diacron (Grosseto, Italy). The oxidative stress index (OSI) was calculated using the ratio ROMs/BAP, as proposed by Ranade et al. [38].

Plasma total polyphenols (PTP) and free polyphenols (PFreeP), both expressed as μg GAE/mL, were determined using the Folin–Ciocalteu method [34], after the extraction procedure described by Serafini et al. [39] for PTP and Santiago-Arteche et al. [40] for PFreeP. Analyses of serum non-esterified fatty acids (NEFA) were performed using the commercial kits FA 115 (Randox Laboratories) according to the manufacturer’s instructions.

### 2.4. Statistical Analysis

Data were analyzed statistically using the MIXED procedure in SAS 9.2 software [41]. For ewes’ live weight, digestibility, and blood parameters, a mixed model was used, with experimental phase (2 levels), parity (P, 2 levels = TL and FL), and diet (D, 5 levels = SHL, DSF2, FSF2, FSF4, and FSFL) as fixed factors and the ewe, or the pen for digestibility, were the random factor used as the error term. Data of ewes’ feed intake and milk production were elaborated using a mixed model for repeated measures with experimental phase, P, D, and the sampling day within the phase (SD, 5 levels for intake and milk yield, 2 levels for milk components) as fixed effects, in which SD represented the unit for repeated measures and the ewe, or the pen for intake data, were the repeated subject, treated as a random factor and used as the error term. Interactions were removed from the models because they were not significant. When the effect of diet resulted as significant (*p* ≤ 0.05), means were compared using *p*-values adjusted according to the Tukey–Kramer multiple comparison test.

## 3. Results and Discussion

### 3.1. Feeding Resources and Feed Intake

Table 1 shows the chemical composition of sulla forage before and after dehydration and pelleting, as well as that of FSF, SH, and the concentrated feed received by the experimental ewes. In the comparison with sulla forage from which it was derived, the DSF shows slight variations in the content of protein and fiber fractions, but a high reduction in CT and total polyphenols, and also in antioxidant capacity, which can be attributed to the heater temperatures of the dehydration and pelleting treatments [25].

Moreover, when DSF is compared to the FSF offered to ewes, comparable levels can be observed for protein, vitamin E, and PUFA, such as linoleic (LA) and α-linoleic (ALA) acids. Because vitamin E was not determined in the pre-dehydration sulla forage, the analogous content detected in DSF and FSF suggests that vitamin E may have had no detrimental effect due to dehydration.

Meanwhile, SH, compared to both FSF and DSF, was characterized by higher fiber components and lower levels in protein, energy, vitamin E, phenolic compounds, and ALA; these differences can be attributed to either the more advanced development stage of the harvested forage, or consistent nutrient loss occurred during the hay-making process. In addition, Rufino-Moya et al. [42] observed an analogous reduction in vitamin E and ALA due to the hay-making of sulla forage, whereas only a slight decrease was recorded in polyphenols and antioxidant activity.

Table 2 reports the results of ewes’ voluntary feed intake and digestibility in relation to the diets. The effects of diet were always significant for the feed intake of forage and the total diet in terms of DM and dietary components. These differences can be mainly related to the higher forage intake recorded with the DSF2 diet, to which the lower dietary proportion of concentrate corresponded (32%), probably due to the smaller forage particle size and the good palatability of the DSF. In contrast to other research [3,4,19], DM intake of diets based on FSF, in which the relative proportion of green forage was equal to 15, 30, and 55% diet DM, was analogous to that recorded with the SHL diet. However, the ewes fed the FSFL diet, with green forage provided ad libitum, showed a superior intake of protein and energy and a lower fiber intake (aNDFom) than the ewes fed the SHL diet.

Compared to the same FSFL, the DSF2 diet induced a higher intake of protein, fiber, and vitamin E, and a comparable energy ingestion. Intake of CT and total polyphenols were lower with the SHL diet, higher with the FSFL diet, and intermediate with the other diets, except for the CT intake with the DSF2 diet, which was analogous to that with SHL. DM digestibility did not differ among diets, whereas digestibility of CT and polyphenols tended towards an increasing trend with the increase of FSF amount in the diet. In relation to parity, the higher ingestions of DM, protein, energy, and vitamin E showed by TL ewes was linked only to the higher amount of concentrate received (44% vs. 35% diet DM), but these differences did not influence the digestive utilization of the diets.

### 3.2. Ewes’ Live Weight and Milk Production

Milk yield (Table 3) with the DSF2 diet was comparable to that obtained with the FSF4 and FSFL diets, characterized by higher incidences of green forage. Moreover, these diets (DSF2, FSF4, and FSFL) increased daily milk yield by more than 100 g in comparison with SHL. This result showed the favorable impact of FSF on milk yield and proves an analogous effect when FSF is replaced by DSF. However, the efficiency of feed utilization for milk production, expressed by the ratio of DM intake to milk yield, resulted lower and more favorable with the FSF4 and FSFL diets, with a higher proportion of green forage.

During the experiment, all ewes increased their live weight, with the exception of those fed the FSFL diet, which showed a slight weight loss; this result can be explained by the higher milk yield of FSFL ewes that, although comparable to that of ewes fed the DSF2 diet, was supported by a lower protein intake.

In relation to milk composition (Table 3), fat was lower in FSFL milk than in SHL milk, presumably as a consequence of the higher milk yield and lower fiber intake of FSFL ewes. Instead, the effect of diet on protein and casein percentages in milk emerged only at level of tendency (*p* < 0.10), which can be related to their slight increase with the FSFL diet in comparison with the SHL diet, whereas urea did not show changes induced by the diet. A worsening in clotting ability was recorded only for SHL milk, due to its curd-firming time (k_20_, min), which was longer than that of FSF2 milk However, the clotting parameters were always comparable among diets containing FSF and DSF.

These results confirm the positive effects of FSF evidenced in previous investigations [3,4,20], especially regarding its high CT content, on the efficiency of dietary protein utilization that, by increasing the availability of amino acids at the udder level, improves milk production [6,11]. However, the effect of CT in increasing the milk casein content emerged only as a tendency with the FSFL diet and was attenuated with the reduction in the amount of FSF in the diet or when FSF was replaced by DSF.

On the whole, the results also demonstrate that DSF was able to improve milk yield similarly to FSF, without affecting the level of milk components.

As expected, parity influenced some traits of milk production; indeed, TL ewes showed higher milk yield by almost 400 g/day, to which corresponded a lower content in protein and casein and a lower level of curd firming time (k_20_, min).

### 3.3. Ewes’ Oxidative and Metabolic Status

Table 4 reports the results of the effects of diet and parity on the oxidative status biomarkers, plasma polyphenols, and NEFA of ewes.

Diet did not affect NEFA, but it did significantly influence ROMs, OSI, and the plasma concentration of free polyphenols. The lower presence of ROMs recorded with the DSF2 diet resulted in a lower value of OSI in comparison with the other diets. Thus, the DSF2 diet was able to induce a better oxidative status in ewes than that recorded in ewes fed FSF. This result was unexpected, because Giorgio et al. [15] observed in goats that sulla fresh forage improved their oxidative status compared to goats fed pelleted alfalfa; this effect was mainly attributed to the different CT content of the two compared legume forages (19.90 g DE/kg DM vs. 0.5 g DE/kg DM in sulla and alfalfa, respectively), because a positive correlation between CT intake and plasma antioxidant capacity in goats was observed [14]. Concerning dry forage, Di Trana et al. [14] also reported that redox balance improved in goats fed sulla fresh forage compared to those fed mixed hay ad libitum plus barley as a supplement.

However, the inconsistency among the results of various studies can be attributed to the complex mechanism of interactions between the substrate ingested by the animal and the ruminal microbial population. In addition, several factors contribute to the final result, such as animal species, interactions with the feed matrix or feed in the animal’s diet, and type of CT, the latter varying according to the plant’s phenological stage, chemical structure, and degree of polymerization [23,43]. In this regard, Giorgio et al. [44] found differences in the antioxidant capacity, polyphenols, and fatty acid content in goat cheeses obtained from animals fed with four fresh forage legume species (*Pisum sativum*, *Trifolium alexandrinum*, *Vicia faba minor*, and *Vicia sativa*).

Certainly, the better oxidative status of ewes obtained with DSF can be a result of the antioxidant protection exerted by the high vitamin E intake, in accordance with Di Trana et al. [45], who observed how feeding regimens based on green forage rich in α-tocopherol can improve the oxidative status of goats. However, the improvement of the oxidative balance of ewes fed a DSF2 diet can also result from the portion of dietary CT metabolites preserved after the dehydration [24,25] and transferred to plasma [22] by a mechanism not yet fully explicated [23]. Accordingly, the level of plasma-free polyphenols in ewes fed DSF was higher than those fed the SHL diet, which provided less CT and polyphenols, but was analogous to those of ewes fed FSF-based diets, which provided higher CT and polyphenol amounts. In this regard, Nitasha Thakur et al. [25] hypothesized an increase in the bioaccessibility of polyphenols due to the dehydration process, which could also explain the higher level of free polyphenols with the DSF2 diet and their contribution in reducing the levels of ROMs and OSI and improving the oxidative balance of ewes.

As reported and discussed in the second part of this study [26], the better oxidative status that emerged for ewes fed DSF corresponded to an improved oxidative stability of derived cheeses, whereas no effect was observed on the cheese antioxidant capacity, which instead improved in cheeses obtained from ewes fed an FSF-based diets.

Parity significantly affected only the plasma concentration of NEFA, which resulted higher in FL ewes; this increase seems to balance the smaller total daily intake of NE_L_ of younger ewes in relation to their milk production level. Both factors, such as energy intake and milk yield, contribute to the energy balance of FL ewes by generating a slight increase in lipomobilization, although the level of NEFA was lower than in those indicating no lipomobilization during the first weeks after delivery (0.20–0.21 mmol/L) [46].

## 4. Conclusions

This research was able to verify the effects of dehydrated and pelleting sulla forage in the diet on milk production and the oxidative status of ewes.

The diet with FSF as an exclusive forage source showed its superiority in terms of milk yield and efficiency of feed utilization for milk production, attributable to its CT content.

Dehydration slightly modified the contents of protein and fiber of FSF and seems to induce no loss of vitamin E and PUFA, but it does reduce CT and polyphenols. Thus, the inclusion of 2 kg of DSF in the diet resulted in higher DM, protein, and vitamin E intake compared to the other diets, but there was analogous energy intake and lower ingestion of CT and polyphenols compared to the FSLS diet. Moreover, milk yield from ewes fed a DSF-based diet improved in comparison with a diet exclusively based on sulla hay, and was comparable to that with diets containing 30 or 55% FSF.

In addition, the positive effect of dehydration of sulla forage on the bioaccessibility of vitamin E and phenolic compounds, exerting antioxidant protection, seems to enhance the oxidative balance of ewes.

These results confirm that the positive effects of sulla also occur when FSF is replaced with DSF. On this basis, the storage and use of sulla dehydrated pellets represent a valid opportunity to exploit the potential of sulla forage in periods of low availability of grazing sources to maintain adequate dairy productions and suggest interesting perspectives in terms of animal welfare and sustainability.

## Figures and Tables

**Table 1 animals-12-02317-t001:** Chemical composition (% DM) and fatty acid content (g/kg DM) of forages and concentrate feed received by ewes.

	Sulla Hay	Pelleted Dehydrated Sulla Forage		Fresh Sulla Forage	Concentrate Feed
		Pre Dehydration	Post Dehydration and Pelleting		
Dry matter (DM), %	90.74	14.23	90.84	18.46	90.90
Crude protein (CP)	7.49	15.88	14.72	14.82	17.71
Ether extract (EE)	1.03	2.40	2.02	2.11	3.26
Ash	9.23	11.85	11.92	11.53	5.93
aNDFom	64.66	48.50	51.20	41.71	19.69
ADFom	52.69	39.54	44.12	33.58	11.36
ADL	10.27	4.57	5.53	7.15	3.28
Cellulose	41.19	34.82	37.88	26.02	7.83
Hemicellulose	11.97	8.96	7.08	8.13	8.33
Non-structural carbohydrates (NSC)	17.59	21.38	20.14	29.84	53.41
NE_L_, kcal/kg DM	695	1101	933	1349	1791
Vitamin E, mg/kg DM	4.96	-	23.63	22.81	15.22
Condensed tannins, g DE/kg DM	2.28	17.91	5.36	27.96	0.76
Polyphenols, g GAE/kg DM	8.29	23.62	11.80	29.40	5.30
TEAC, mmol trolox/kg DM	57.28	122.75	87.47	135.42	36.07
Lauric acid, C12:0	0.048		0.066	0.086	0.027
Myristic acid, C14:0	0.11		0.078	0.10	0.13
Palmitic acid, C16:0	2.31		2.84	3.04	7.06
Stearic acid, C18:0	0.44		0.51	0.68	0.75
Oleic acid (OLA), C18:1 c9	0.85		0.63	0.82	7.43
Linoleic acid (LA), C18:2 n − 6	1.13		2.00	1.86	8.05
α-linolenic acid (ALA), C18:3 n − 3	0.54		7.04	6.24	0.51

aNDFom = neutral detergent fiber using heat-stable amylase and exclusive of residual ash. ADFom = ash free acid detergent fiber. ADL = acid detergent lignin. NSC% = 100 − (CP% + EE% + ash% + aNDFom%). NE_L_ = net energy for lactation according to INRA (2018). DE = delphinidin equivalent. GAE = gallic acid equivalent. TEAC = trolox equivalent antioxidant capacity.

**Table 2 animals-12-02317-t002:** Effect of diet and parity on voluntary feed intake and digestibility.

	Diet (D)						Parity (P)			Significance *p* < (1)	
	SHL	DSF2	FSF2	FSF4	FSFL	SEM	TL	FL	SEM	D	*p*
**Daily forage intake**											
Dry matter (DM), g	1442 b	1890 a	1524 ab	1392 b	1152 b	107.07	1500	1495	67.72	0.0002	0.6686
Crude protein (CP), g	121 c	260 a	150 c	155 c	208 b	9.82	179	179	7.30	<0.0001	0.9881
aNDFom, g	862 a	804 a	809 a	695 ab	417 b	76.90	716	719	50.63	0.0004	0.9743
NE_L_, kcal	1086 b	2006 a	1448 b	1221 b	2132 a	133.13	1604	1553	144.76	<0.0001	0.8028
Vitamin E, mg	9.70 d	41.39 a	16.21 c	20.54 c	30.84 b	1.19	23.93	23.55	1.00	<0.0001	0.7897
Condensed tannins, g DE	7.32 d	8.09 d	10.99 c	23.10 b	42.35 a	1.52	18.52	18.22	1.91	<0.0001	0.9107
Polyphenols, g GAE	17.78 c	28.19 b	23.96 b	28.36 b	45.42 a	2.10	28.95	28.53	2.45	<0.0001	0.9034
**Total daily feed intake**											
Dry matter (DM), g	2351 b	2799 a	2433 ab	2301 b	2061 b	107.07	2591	2186	67.72	0.0002	<0.0001
Crude protein (CP), g	282 c	421 a	311 c	316 c	369 b	9.82	372	308	7.29	<0.0001	<0.0001
aNDFom, g	1041 a	983 a	988 a	874 ab	596 b	76.90	931	862	50.63	0.0004	0.3360
NE_L_, kcal	2714 b	3634 a	3076 b	2849 b	3761 a	133.13	3558	2855	144.75	<0.0001	0.0009
Vitamin E, mg	23.53 d	55.22 a	30.05 c	34.37 c	44.67 b	1.93	40.52	34.61	1.00	<0.0001	<0.0001
Condensed tannins, g DE	8.02 d	8.78 d	11.68 c	23.79 b	43.04 a	1.52	19.36	18.7	1.91	<0.0001	0.8303
Polyphenols, g GAE	22.60 c	33.01 b	28.78 b	33.18 b	50.24 a	2.10	34.73	32.38	2.45	<0.0001	0.4995
**Digestibility, %**											
Dry matter (DM)	81.39	81.87	81.19	80.38	82.48	3.75	82.65	80.27	2.37	0.9959	0.4892
Crude protein (CP)	75.05	79.60	77.13	75.72	79.39	4.79	79.08	75.67	3.03	0.9402	0.4401
aNDFom	75.81	69.25	67.88	68.84	69.79	6.20	70.05	70.58	3.92	0.8997	0.9247
NE_L_	81.37	81.97	80.68	80.22	82.28	3.78	82.41	80.20	2.39	0.9943	0.5240
Condensed tannins	87.68 a	90.95 a	95.06 bc	96.84 bc	98.32 c	1.57	93.82	93.72	0.99	0.0019	0.9488
Polyphenols	82.74	89.60	89.89	92.77	94.92	2.75	90.08	89.89	1.74	0.0701	0.9419
Concentrate, % diet DM	41.44 a	32.41 b	38.46 a	40.18 a	44.20 a	1.52	43.66	35.0	0.96	<0.0001	<0.0001
Fresh sulla forage, % diet DM	-	-	15.06 c	30.44 b	55.26	1.40	30.69	36.48	1.24	<0.0001	0.0018

SHL = Sulla hay ad libitum. DSF2 = 2 kg/day dehydrated sulla forage. FSF2 = 2 kg/day of fresh sulla forage. FSF4 = 4 kg/day of fresh sulla forage. FSFL = fresh sulla forage ad libitum. TL = third-lambing ewes. FL = first-lambing ewes. aNDFom = neutral detergent fiber using heat-stable amylase and exclusive of residual ash. NE_L_ = net energy for lactation according to INRA (2018). DE= delphinidin equivalent. GAE = gallic acid equivalent. SEM = standard error of mean. (1) On the row: a, b, c, d = *p* < 0.05.

**Table 3 animals-12-02317-t003:** Effect of diet and parity on live weight variation and individual milk yield and composition.

	Diet (D)						Parity (P)			Significance *p* < (1)	
	SHL	DSF2	FSF2	FSF4	FSFL	SEM	TL	FL	SEM	D	*p*
Initial live weight, kg	50.27	49.41	47.35	49.96	49.09	1.32	52.69	45.74	1.48	0.2905	0.0046
Final live weight, kg	50.65 a	50.96 a	49.97 ab	51.45 a	48.86 b	1.18	53.62	47.14	1.59	0.0105	0.0115
Weight gain, kg	0.35 ab	1.91 ab	2.61 a	1.53 ab	−0.59 b	0.70	0.93	1.40	0.44	0.0396	0.4708
Milk yield, g/day	1781 b	1914 a	1780 b	1872 ab	1925 a	77.23	2046	1662	93.75	0.0029	0.0042
DM intake/milk yield, kg/kg	1.38 a	1.35 a	1.33 ab	1.17 bc	1.15 c	0.063	1.26	1.29	0.041	0.0170	0.5626
Lactose, %	4.56	4.52	4.59	4.55	4.59	0.041	4.60	4.53	0.047	0.2693	0.3265
Fat, %	5.34 a	5.24 ab	5.07 ab	5.28 ab	4.76 b	0.18	5.11	5.17	0.19	0.0310	0.8149
Protein, %	4.62	4.73	4.70	4.77	4.82	0.095	4.54	4.91	0.11	0.0821	0.0172
Casein, %	3.40	3.51	3.46	3.54	3.57	0.083	3.34	3.66	0.095	0.0767	0.0206
Lactose, g/day	73.55 b	84.79 a	79.92 ab	79.30 ab	84.07 a	3.74	91.38	69.27	4.43	0.0017	0.0009
Fat, g/day	85.14 b	98.54 a	87.93 b	92.55 ab	86.83 b	4.96	100.31	80.09	5.24	0.1240	0.0082
Protein, g/day	73.81 b	88.73 a	81.51 ab	82.72 a	86.80 a	3.88	90.00	75.42	4.49	0.0001	0.0255
Casein, g/day	54.25 b	65.97 a	60.06 ab	61.24 a	64.11 a	2.94	66.05	56.20	3.37	0.0001	0.0436
Urea, mg/dl	27.22	27.25	30.77	29.48	27.79	1.84	30.42	26.59	1.72	0.2578	0.1208
Coagulation time (r), min	17.68	17.27	16.52	17.99	18.94	1.01	18.32	17.04	1.10	0.1213	0.4146
Curd firming time (k_20_), min	1.90 a	1.74 ab	1.37 b	1.61 ab	1.78 ab	0.13	1.87	1.49	0.11	0.0084	0.0217
Curd firmness (a_30_), mm	50.56	55.04	57.64	52.12	53.47	2.36	50.95	56.58	2.13	0.2046	0.0679
Curd firmness (a_2r_), mm	54.81	59.39	60.99	58.28	59.21	1.75	56.91	60.16	1.26	0.1048	0.0771

SHL = sulla hay (SH) ad libitum. DSF2 = 2 kg/day dehydrated sulla forage (DSF) plus SH ad libitum. FSF2 = 2 kg/day fresh sulla forage (FSF) plus SH ad libitum. FSF4 = 4 kg/day FSF plus SH ad libitum. FSFL = FSF ad libitum. TL = third-lambing ewes. FL = first-lambing ewes. DM = dry matter. SEM = standard error of mean. On the row: a, b, c = *p* < 0.05.

**Table 4 animals-12-02317-t004:** Effect of diet and parity on oxidative status biomarkers, plasma polyphenols, and NEFA in ewes.

	Diet (D)						Parity (P)			Significance *p*< (1)	
	SHL	DSF2	FSF2	FSF4	FSFL	SEM	TL	FL	SEM	D	*p*
Reactive Oxigen metabolites (ROMs), Unit Carr	79.68 ab	72.47 b	82.96 a	86.39 a	84.22 a	5.13	79.14	83.15	5.47	0.0462	0.6073
Biological antioxidant potential (BAP), mmol/l	2.13	2.51	2.21	2.19	2.31	0.18	2.04	2.50	0.22	0.1526	0.1473
Oxidative Stress Index (OSI = ROMs/BAP)	40.77 a	30.4 b	41.21 a	42.09 a	37.54 ab	3.46	41.05	35.78	3.47	0.0003	0.3661
Plasma Total Polyphenols (PTP), μg GAE/ml	18.09	18.40	18.12	17.25	18.23	0.42	17.65	18.39	0.34	0.1602	0.1396
Plasma Free Polyphenols (PFreeP), μg GAE/ml	11.44 b	13.13 a	12.52 ab	12.61 a	12.63 a	0.46	12.46	12.47	0.45	0.0335	0.9862
NEFA, mmol/l	0.093	0.090	0.111	0.109	0.084	0.015	0.078	0.117	0.011	0.5258	0.0113

SHL = sulla hay (SH) ad libitum. DSF2 = 2 kg/day dehydrated sulla forage (DSF) plus SH ad libitum. FSF2 = 2 kg/day fresh sulla forage (FSF) plus SH ad libitum. FSF4 = 4 kg/day FSF plus SH ad libitum. FSFL = FSF ad libitum. TL = third-lambing ewes. FL = first-lambing ewes. NEFA = non-esterified fatty acids. GAE = gallic acid equivalent. SEM = standard error of mean. (1) On the row: a, b = *p* < 0.05.

## Data Availability

Not applicable.

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
