# Peer review of "Feeding Dairy Ewes with Fresh or Dehydrated Sulla (Sulla coronarium L.) Forage. 1. Effects on Feed Utilization, Milk Production, and Oxidative Status"

_animals, 2022, doi:10.3390/ani12182317_

Round 1
Reviewer 1 Report
General comments:
The document entitled "Feeding dairy ewes with fresh or dehydrated sulla (Sulla coronarium L.) forage. 1. Effects on feed utilization, milk production and oxidative status" is well written and does have merit worth publication. There are minor correction to be made to the English (Simple summary, abstract and introduction). The use of alternative feeding materials in ruminant production systems is need (esp. if it can reduce GHG emission, improve productivity and improve animal products).
The results and discussion section should be separated. This will give better clarity to both sections for the reader. In the discussion section, more information can be given the level of antioxidants in blood and milk (animals products) with respect to the processing of forages (dried, pelleted, fresh) of various forage legume species. This will give more evidence based on the concluded statement on the increased bioavailable compound in pelleted forage legumes.
Specific comments:
In the abstract, line 32, is this a typo? Should the acronym read DSF?
Line 44: please revise (improve clarity in statement).
The scientific name of the forage should be italicized throughout the text.
Line 364: please revise "to counter balance"
Institutional Review Board Statement: The last paragraph of section 2.2 should also be placed here.
Author Response
"Please see the attachment."

Reviewer 2 Report
The work is original and provides very interesting background on the use of local resources that are a real alternative for a sustainable animal production.
Author Response
REVIEWER 2
The work is original and provides very interesting background on the use of local resources that are a real alternative for a sustainable animal production.
AU: Thank you very much for these positive considerations.
Reviewer 3 Report
The subject of the manuscript is Feeding dairy ewes with fresh or dehydrated on the (Sulla coronarium L.) forage. 1. Effects on feed utilization, milk production and oxidative status.
The study is carried out correctly even if the subject has been treated extensively by the authors themselves, as can be seen from the list of references. Furthermore, English is not easy to read and must be carefully revised.
The introduction is too long and dispersed, it must be shortened and centered. The materials and methods are correct and the analyzes are appropriate. Unclear how many plasma samples were analyzed.
In the abstract and in other parts of the manuscript , a DSF group that is not present in the M&M, presumably refers to the DSF-2 group, be standardized. The conclusions talk about an increase in the quantity of proteins and caseins in milk. This fact is only the result of the increase in daily milk production and as such the data must be treated.
Too many self-citations and some unimportant for the present study.
The study must be thoroughly reviewed and the data obtained must be presented for the proper value they have.
Kind regards
Author Response
"Please see the attachment."

Round 2
Reviewer 1 Report
All corrections were made. The rebuttal to keep the structure of results and discussion was properly justified.
Paper should be published as is.
Reviewer 3 Report
Dear Ms. Vega Liu
Assixtent Editor of Animals
The manuscript has been improved and made easier to understand. In this form the manuscript, in my humble opinion, can be published in the journal.
Kind regards
Vincenzo Carcangiu